# Usability Tests for Texture Comparison in an Electroadhesion-Based Haptic Device

**Afonso Castiço** [1,*] **and Paulo Cardoso** [2]

1    Department of Electrical and Computer Engineering (DEEC), University of Coimbra,
     3030-290 Coimbra, Portugal
2    Department of Industrial Electronics (DEI), University of Minho, 4800-058 Guimarães, Portugal
*    Correspondence: uc2016223556@student.uc.pt

**Abstract:** Haptic displays have been gaining more relevance over the recent years, in part because of the multiple advantages they present compared with standard displays, especially for improved user experience and their many different fields of application. Among the various haptic technologies, electroadhesion is seen as capable of better interaction with a user, through a display. TanvasTouch is an economically competitive haptic device using electroadhesion, providing an API and respective haptic engine, which makes the development of applications much easier and more systematic than in the past, back when the creation of these haptic solutions required a greater amount of work and resulted in ad-hoc solutions. Despite these advantages, it is important to access its ability to describe textures in a way understandable by the user's touch. The current paper presents a set of experiments using TanvasTouch electroadhesion-based haptic technology to access how a texture created on a TanvasTouch device can be perceived as a representation of a real-world object.

**Keywords:** usability tests; haptic; electroadhesion; TanvasTouch; haptic touchscreens; user experience; texture comparison

## 1. Introduction

The word "haptics" concerns everything related to the sense of touch, which means that haptic technology is a specific branch of technology that recreates artificial tactile stimuli that can be perceived by average users.

Besides having traditional visual and audible channels of interaction, haptic displays also provide tactile stimulation that extends the capabilities of standard touchscreens. Using haptic displays positively contributes to deeper and more immersive user experience (UX) interactions, considerably extending the number of application areas where this type of technology can be used [1]: from the automotive industry to consumer electronics, from retail to gaming, or even digital signature based on touch, there are many fields where haptic technology can be useful. Moreover, new innovative solutions can be designed based on this type of technology to mitigate visual impairment and other possible disabilities, creating much more inclusive technology.

Regarding haptic technology, there are three main different types of displays: electrostatic, vibrotactile, and ultrasonic haptic devices. This paper is focused on electroadhesion, a type of electrostatics.

Electrostatic displays owe their name to the electrostatic physical principle with the same name. When it comes to the physics that sustain electrostatic displays, this type of technology is based on friction modulation between the human finger and the touchscreen, according to Nakamura and Yamamoto [2]. By changing the voltage difference between them, it is possible to vary the friction force on the surface, which is perceived by the user when touching the haptic touchscreen. For this to happen, a thin metal layer is required above the display, which reveals a characteristic sensation under the application of AC

voltage to that layer. When a human finger gets close to this surface, the electrostatic force between the finger and the layer acts due to the voltage difference between the two. When the finger slides over the surface, the electrostatic force is converted into friction, which is experienced by users as haptic texture.

As seen, electrostatic haptic displays benefit from the resulting electrical properties between human skin and a charged surface, in this case a touchscreen display [2]. According to [3], electrostatic actuation is highly effective in increasing the friction in a touchscreen, once the application of voltage to a conductive layer increases the electrostatic attractive force in a perpendicular direction to the surface, which means that haptic textures displayed on these touchscreens are better perceived by users. This tactile effect felt by a user can be changed and modulated by varying the amplitude, frequency, or waveform of the voltage that is being applied to the touchscreen.

Even though electroadhesion is electrostatic-based haptics, special focus is given to this principle because it explains how the TanvasTouch (TT) device works from a technical perspective. The electroadhesion principle is also known as the electric- based adhesion effect. It consists of a local variation of electric fields where fingers interact with the touchscreen [4]. Another way to describe this effect is as a modulation of friction between the human fingertip and an active surface, according to Soft Matter journal [5]. The glass plate's insulating layer and the human fingertip are inductively polarized, which in practice means that charges with the opposite sign are progressively accumulated in each contact surface: the positive charges from human skin and the negative charges on the insulating layer. As known, opposite charges attract each other and when the human fingers slip over the haptic touchscreen, these opposite charges crash between each other, originating friction, which is perceived by users as texture on the haptic touchscreen.

Another type of display is the vibrotactile displays. These ones consist of the mechanical stress of haptic actuators. The main actuators used in these displays are eccentric rotating mass actuators (ERM), linear resonant actuators (LRA), and piezoelectric actuators. ERM and LRA actuators are commonly applied for the same purposes [6,7], mainly used in smartphones. From study [8], it was concluded that LRA actuators are much more battery efficient than ERM actuators. Piezo actuators have also been recently embedded on the new Windows 11 trackpads, according to Boréas Technologies [9].

There are two possible approaches to the design of haptic textures with this type of technology [10]: monolithic vibrotactile displays, which correspond to vibrating an entire rigid display, and a localized vibrotactile display approach, where several haptic actuators are integrated on the displays to promote vibrating stimuli on localized areas of the touchscreen or object. Some of the main advantages of this type of haptic technology are its high customization and goal- orientated personalization according to the desired purpose and haptic effect. Moreover, they can deliver a better UX once the design of these displays is much more flexible to adapt to specific user needs. The drawbacks of this technology are mainly the lack of robustness since the final products might not be as compact as the haptic displays assembled on a single component. There is also an increment in design complexity since the engineering of the whole device from scratch can be particularly challenging. Finally, due to their ability to be customized and personalized, vibrotactile technology has been extended beyond touchscreens. Many different applications can be found based on vibrotactile haptics, mainly wearables, internet of things applications, gaming, remote controllers, haptic gloves, or armbands [10]. From [11], vibrotactile feedback has been proven to be very effective to distinguish between different texture patterns.

Finally, ultrasonic displays. According to Wilson and Carter et al. [12], ultrasonic-based haptic solutions consist of the modulation of air pressure waves from a display of physical ultrasound transducers, instead of the previous haptic solutions that were mentioned before. Ultrasonic haptics can reproduce haptic stimuli without the need to call on modulation of physical friction or electronic enginery seen previously with electrostatic and vibrotactile displays. According to Sun and Nai et al. [13], the ultrasound transducers from the display generate a pulse that reaches a coincident focus point in the middle of

air at the same time since all the pulses have the same phase at the targeted point. This process is known as ultrasound focusing. When a hand is positioned above the focus point, a tactile sensation can be experienced in 3D space. Having this set and applying the same logic for multiple points in three-dimensional space, it is possible to produce 3D tactile objects that are perceivable by human skin. Some of the advantages related to ultrasonic haptics are its extended haptic-based friction modulation without the need to use physical interfaces and the ability to represent haptic textures on three dimensions, which is a major advantage compared to electrostatic and vibrotactile displays [3]. The main disadvantages of ultrasonic haptic technology are the resonant nature of ultrasonic waves that introduce undesired noise and require an extra effort to cancel. Furthermore, it is important to keep a balance between energy efficiency and the amount of bandwidth required for the desired purposes, resulting in an extra concern that may limit the usage and applications of this type of haptic technology. The biggest potential of this technology has not yet been reached and there is a considerable number of different applications of ultrasonic technology that can be explored even further in the future, according to Rakkolainen and Sand et al. [14], namely virtual reality (VR), augmented reality (AR) applied to gaming scenarios and simulated environments, buttonless interfaces for the automotive industry, telemedicine, and remote surgeries, etc. The main obstacles that ultrasonic haptics technology is facing are the reduced magnitude and limited reachability of ultrasonic- related haptic applications, the weight and size of the device, and the high pricing.

To understand the aim and motivation of this paper, it is important to acknowledge that from the five existing senses of human beings, touch has been one of the less explored by technology. This field has evolved very slowly over the years, probably because it is difficult to artificially reproduce touch, due to the subjective and complex nature of this sense. It is easy to understand why visual and hearing-based technologies were developed first, as they require less sensitive interaction from the user, and hence they are easier to implement.

This article is motivated by the opportunity to explore emergent and innovative haptic technology that contributes with applied knowledge to the field of human–machine haptic interfaces. Keeping this top of mind, this article presents UX tests based on electroadhesion haptic technology, which supports the TT device used in these UX tests. Being one of the few technologies commercially available using electroadhesion, there are not many studies on this performed in the field.

In the following section, different articles are cited related to other UX tests that also use the same TT device in their experiments and some other papers that explore other haptic technologies.

## 2. Literature Review

The automotive industry is one of the best examples to understand how crucial and valuable the haptic devices can be. Nowadays, technology is becoming more and more present in our lives, and traditional vehicles have been following this current and future trend of embedding more and more haptic technology in the cockpit. With a blink of an eye, vehicles are becoming the second place where humans spend more time, after their own homes. Since 2008, cars have increasingly been equipped with different infotainment devices with a wide range of functionalities, mostly based on touchscreen technologies that do not offer real time feedback when a user interacts with them, which can be translated in a high visual workload that can distract drivers' attention and represent risk when it comes to human safety when driving [15]. One of the weaknesses of conventional touchscreens is that they demand considerable attention from user's visual attention, which in these contexts is problematic and distracts the automobilist from his main task of driving. In this sense, haptic environments intend to bring a tactile context to facilitate this operation, minimizing visual interaction.

To better understand the problem of the increasingly visual workload on standard touchscreen displays, different studies [15,16] have been performed to better quantify the

influence of these technologies on the performance of drivers. The first study [15] consisted in a "search and select" type task, where each volunteer was asked to search for a specific button in an array of different buttons on the vehicle's touchscreen, while driving in a realistic driving simulator, to avoid possible damages if these dangerous activities were performed in a real highway. Each driver performed two times this exercise, first with the support of "Visual Only" touchscreen feedback, then repeating the same exercise but this time with a combination of "Visual+Haptic" feedback. This article highlights how important the combination of "Visual+Haptic" feedback is to reduce the number of glances required to perform in-vehicle touchscreen operations. With "Visual Only" feedback, most users required two glances to perform the required task. When compared with a combination of "Visual+Haptic" feedback, most users were able to execute similar tasks with a one-time glance performance, which represents an important time saving benefit when it comes to driving.

A second study [16] was led to evaluate the combination of multimodal stimulus (audio (A), visual (V) and tactile/haptic (T)) and better understand their influence in driver's response time in different driving contexts like "Car Braking and Warning", "Car Braking Only", and "Warning Only". This study concluded that a combination of different modalities, bimodal (AV, AT, and TV) and trimodal (ATV), represents a significant decrease in the drivers' response time when compared with individual unimodal modalities, which require a higher brain processing effort due to the lack of more intuitive feedback that can be obtain by the combination of audio, visual and tactile feedback. Paper [17] concluded that increasing the haptic feedback, besides the dashboard, in the steering wheel, seat, seat belt, and driver's clothes, also decreases the number of hazards on the road.

Another article tested the HapTouch system [18], studying the use of haptic feedback in touchscreens and its effect in security hardening. HapTouch consists of an in-vehicle touch-based interface with tactile feedback, pressure and force-sensitive to human touch, with embedded sensitive displays, bearings, and voice actuators. This technology is particularly innovative by adding an extra state model to the standard state space models that are traditional in this type of touch screen devices.

The simple space model shown on the original article represents the interaction of a user with the touch screen panel: when the finger is not directly touching the interface, "Passive tracking" loop is occurring infinitely in State 0. When "Contact" happens, this enables a transition to State 2, to allow a "Selection" made by the user, when interacting with the touchscreen surface. The action of releasing the finger, "Release Contact", allows the system to return to its original state, State 0. There is no "State 1" to better introduce the innovation brought by the HapTouch. The device adds an extra state, State 2 + 1, that results from a combination of State 1 and State 2. This modification allows a continuous movement by dragging the finger on the surface, from a starting point to an ending point. The input signal feed to the system is directly proportional to the amount of pressure made on the HapTouch surface. More details about the mentioned space models provided in the original article [18]. Another conclusion reached with this study is a significant error reduction related with tasks that involve the selection of numeric buttons on the screen, which brings a considerable security improvement.

A literature review article [19] tested up to seventy different current haptic solutions and studies applied or related with touchscreen driving technologies and concluded that warning systems are more frequently communicated to the driver with the usage of vibration/haptic solutions, meanwhile guidance systems normally use force feedback to update the driver with road alerts. Overall, haptic feedback offers improved performance, lower reaction times, and lower mental effort to perform the required tasks, which translates into higher security for the driver and the vehicle occupants.

This short introduction allows the reader to understand how haptic technology can be useful applied to specific industries such as automotive.

One of the main articles consulted was from Park et al. [20], which was focused on understanding how tactile feedback can influence a user's preference under evaluation of

2D images. Regarding the obtained results, it became clear that volunteers prefer images that they could feel (images with a haptic texture added to the visual stimuli were preferred to the ones without haptic textures). The sharp tactile texture was the most popular one due to the high definition of its haptics compared to the blurred or mismatched ones. The authors concluded that the quality of the haptic texture was essential to the user's experience and their preference in haptic feedback experiments. Paper [21] concluded that the larger the period of real texture is, the easier it is for users to recognize its pattern and periodicity. Hence, the accuracy of virtual textures is prone to be lower than real textures.

Another important experiment was conducted concerning textures renderization using a TT device. The article from Klatzky and Nayak et al. [22] contributed to understanding the identification and matching process between haptic textures and patterns and the correspondent visual images. This paper focused its attention on understanding how users detect and identify tactile information when dealing with haptic technology. The authors of this study used the TT device to create two different exercises: the first one concerning the detection of friction change on haptic textures, and the second one related to matching haptic textures to visual images from where these haptic textures were created from. Between other results, this study concluded that there was a great difference between the ability of users to detect haptic patterns and their ability to match these same haptic patterns to visual images from where they were generated from.

In another publication, Breitschaft and Carbon [23] presented some of the most relevant recommendations that were followed in the implementations of these UX tests. The authors of this article, in partnership with BMW Group, designed two user interfaces (UI) with TT equipment and studied the reaction of participants to electrostatic friction modulation in a UI research environment. Two different exercises were executed. The first one was based on a single search task using low and high frequency textures. The second exercise consisted of a target selection task performed in a driving scenario in a simulated environment. The conclusions reached with this experiment allowed to establish general guidelines for the design of different haptic UI. The suggested guidelines for haptic UI design are the following: (1) Use analogies: by creating different analogies and associations with reality, it is possible to create clear and tangible feedback with the user. (2) Keep it simple: design a simple and basic set of differentiated haptic sensibilities. (3) Make it strong: a set of solid haptic sensibilities transmits better feedback and avoids misunderstandings and false interpretations by the user. (4) Consider Habituation: allowing participants to get comfortable with haptic technology is halfway to a better UX.

Beheshti et al. [24] explore haptic feedback as an education tool to better explain complex and abstract concepts which are normally difficult for young children to understand. An example of this is the functioning principle of electric current flowing on a circuit. To do so, this case study targets parent–child duos invited to execute different tasks related to the selected topic. Furthermore, the haptic feedback solution described within this paper presents useful information concerning the UX and its translation into the haptic design, which is considered useful for the UX tests presented next. Bateman et al. [25] also conducted a usability study presenting a deeper understanding of the importance of designing UX-oriented (user-centered design) haptic solutions for visually impaired people, an audience that highly benefits from haptic technology.

## 3. Materials and Methods

The purpose of this work is to access how users perceive electroadhesion-based haptic technology when interacting with designed textures, i.e., how the device can mimic the perception of real-world objects.

### 3.1. Materials

A TanvasTouch (TT) device was used. TT is a haptic touchscreen that can generate software-based haptic textures. Unlike other haptic devices and technologies, TT is the first commercial ready-to-buy technology that allows the creation of haptic effects on an

API-based environment. When compared with standard haptic actuators (ERM, LRA, and piezoelectric actuators), TT technology allows performing screen interactions without the need for surface vibration, which is the basic functioning principle of other haptic technology, like vibrotactile. Moreover, with a continuous movement of fingers along with the TT touchscreen, it is possible to execute almost every required task on the display, allowing a smoother UX. In the creation of haptic applications using TT, all possible designed solutions require the creation of a graphical user interface (GUI) and the design of an underlying haptic layer that is responsible for adding haptic effects to the elements of the GUI. Only the GUI is presented to the user; the haptic layer is never visible.

There are two main components of this architecture [26]: the personal computer of the user and the TT device itself. Any haptic application (haptic app) created with this device requires adopting a specific TT API [27] (NET API, C API, or C++ API. The NET API was used with VisualStudio IDE) that directly communicates with the TT Engine. This Engine uses a USB cable to exchange friction- related data between the computer and the TT Controller, which is responsible for changing the friction of the haptic display. In more detail, the TT Engine is a software driver that translates the functions of the APIs into practical commands that can be interpreted by the TT Controller to vary the friction of the screen (Friction Surface), according to the code developed in each haptic app. While the friction of the touchscreen is changing, it is necessary to perform real- time updates of the image displayed on the TT screen, which is done with an HDMI cable that directly connects the computer to the Tanvas device.

Regarding the technical characteristics of TT, the video resolution is 1280 × 800, contrast is 800:1, reporting rate is 250 Hz, supported hardware is ×86 (64 bit), Operating System is required to be Windows, power consumption is 10 W max, accuracy is 1 mm in the center area of the touchscreen, and multi-touch is supported up to 10 fingers, according to [28].

Regarding competition, there are still no direct competitors presenting a commercial product with an open source API software based environment that allows the development of haptic applications. Nevertheless, TouchSense [29] and Senseg FeelScreen [30] can be considered as possible competitors.

As seen in the beginning of this paper, one of the main areas of applications of TT technology is mainly the automotive industry [31]. This type of technology can be particularly interesting when applied to ecommerce and online shopping, allowing customers to perceive the textures, having a tactile experience of clothes before paying for them [28]. There are many other examples of application in this field: assisted learning, consumer electronics, gaming, and between many other possible applications of this technology, once haptic feedback technology is extremely customizable to different products and applications.

This work not only follows the guidelines related to the design of the UX haptic tests from [20] and the other mentioned articles, but also introduces some new tests where the haptic textures designed on the electroadhesion-based haptic device are compared with different real-world objects physically presented to the volunteer. Klatzky et al. [22] were influential in this work to understand the identification and matching process between haptic textures and patterns and the correspondent visual images too.

For the tests, 20 volunteers were selected, with no previous experience with any haptic technology. No gender or age aggregation was made as the number of individuals in some sets could be not significant for discrimination of results.

To avoid visual bias, there were used "blind tests" in the sense that the visual stimuli interface of the touchscreen will be the same for all tests. The haptic sensibility of the touchscreen (haptic texture) will change from test to test, which means that the volunteer can only identify the textures when touching the screen.

The purpose of the experiment is to identify haptic textures on TT that match four physical objects during the different tests. To allow the design of software-based textures presented to the volunteers of this experiment, a TT electroadhesion-based haptic touchscreen device was used. The four different objects used for texture comparison are (a) the

interior of an embossed cardboard box, (b) a cork stopper, (c) the surface of different small tiles, and (d) a phone case, as presented in Figure 1.

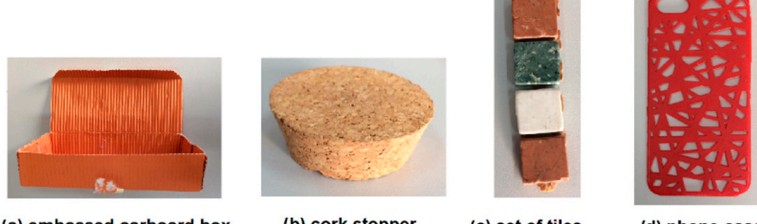

**Figure 1.** The objects used in the UX experiment.

The article from Mun et al. [32] was considered to better select the physical objects used for these UX tests. The authors from this study concluded that the haptic textures that are better perceived by users on a TT device are the ones that have a 'rough-smooth', 'dense-sparse', or 'bumpy-even' texture contrast. The selected objects from Figure 1 were selected keeping in mind these same three texture contrast adjective duos.

*3.2. Methods*

The visual interface presented to the user consists of a single panel divided into four smaller canvasses where different textures are shown in each of the tests, as shown in Figure 2.

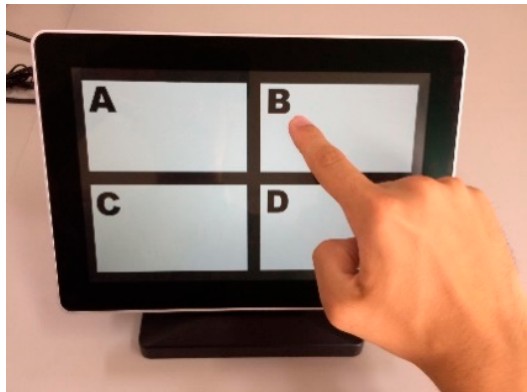

**Figure 2.** Example of user interaction with the TT device.

It is important to access what is meant for "physical textures" (also mentioned as "real textures") and "virtual textures". The first concept corresponds to the textures of the real-world objects presented in Figure 1. The second one corresponds to the textures designed on the TT device that try to mimic and artificially reproduce the ones from the first group.

For each test, the goal is to identify the real texture from the object or objects presented physically to the volunteer based on the identification of the virtual textures presented blindly on TT, as shown in Figure 3.

The blind test set includes five different tests. The first test consists of the univocal matching of all four physical textures from Figure 1 to the corresponding four virtual ones represented on the haptic touchscreen as presented in Figure 3. Figure 4 shows an example of the visual stimuli and the tactile texture presented on Test 1. Tests 2–5 follow similar combinations, as shown in Figure 5.

The remaining tests 2–5 consist of the identification of only one of the physical textures based on the virtual textures reproduced on the haptic touchscreen. This time, all virtual textures on the touchscreen correspond to just one physical object from Figure 1, but

with slight variations of intensity, using blurred image filters, sharp filters, or variation of textures' shape, e.g., mismatched textures or even the complete removal of texture [20], which correspond to the small black canvasses presented in Figure 5.

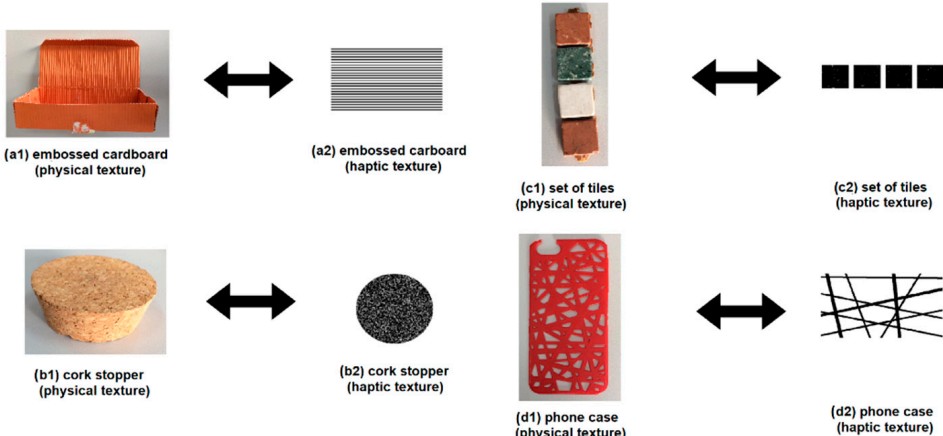

**Figure 3.** Correspondence between physical textures from real-world objects and haptic textures.

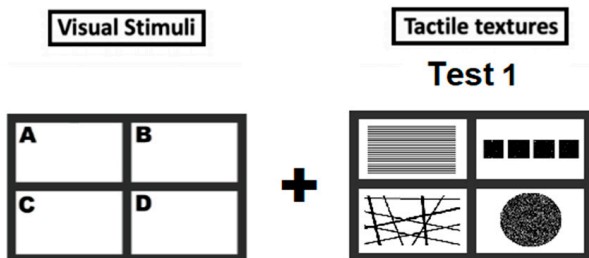

**Figure 4.** Visual stimuli and tactile texture for test 1. Partially adapted from [20].

All the last four tests use the same filters mentioned before but are presented in a different order at a time. In none of the tests was the physical texture that was being evaluated mentioned. Keeping this in mind, test 2 presents texture variations of the cork texture, tests 3 presents variations of the phone case texture, test 4 corresponds to the cardboard texture, and test 5 corresponds to the tiles texture. Figure 5 presents all the five tests at once.

Participants of this experiment were allowed to touch and handle the textures from real-world objects freely during the whole experiment. Participants had the possibility of not choosing any correspondence between the physical and haptic textures in case they considered there was not a reasonable choice to be considered in the set of available haptic textures displayed on the TT screen. Moreover, volunteers could give the same answer in different tests, which means that if they considered that the same virtual texture was being reproduced in different tests, this was considered a valid answer. Finally, the order of tactile textures for each test was defined only one time in random order and kept unchanged during the execution of the blind tests for all volunteers.

Each test was executed only once. Each volunteer decided independently the amount of time required for the execution of each test. Each volunteer had first contact with haptic technology before the experiment started by interacting with the Tanvas Intro App designed by Tanvas. This was a crucial preparation step for the experiment that turned out to be extremely important for the users to get comfortable with this type of haptic technology.

During the execution of each test, the volunteers of this experiment answered some questions from a survey that allowed to obtain the results and conclusions presented in the next section. Please look at Appendix A for more details about the survey.

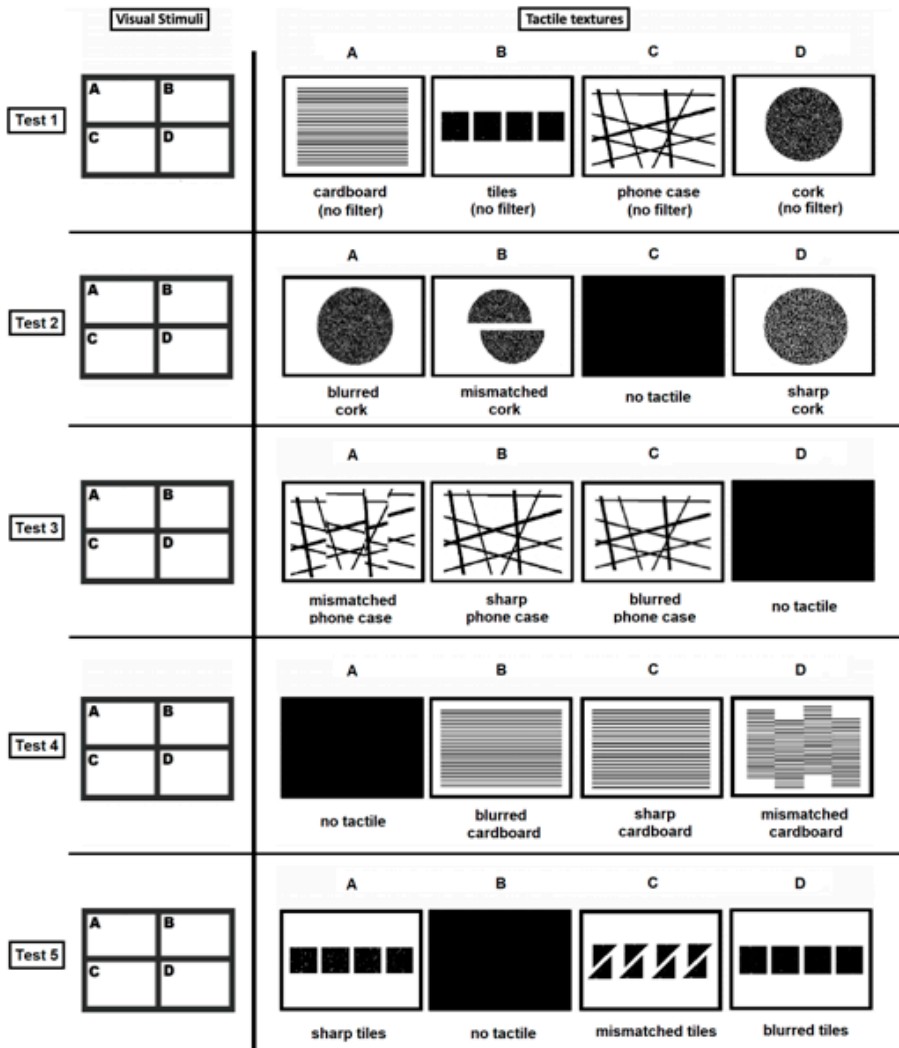

**Figure 5.** Visual stimuli and tactile texture for tests 1, 2, 3, 4 and 5. Partially adapted from [20].

## 4. Results

As explained in the previous section, the goal was to access how users perceive electroadhesion-based haptic technology when interacting with designed textures.

The first test was designed to access if very different textures can be perceived as matching physical objects, providing a first insight concerning the performance of the device, which will be later compared to more detailed perception tests, in tests 2–5. Recalling Figure 4, the setup of this test is presented in Table 1, showing the correct match between the virtual and the physical textures.

**Table 1.** Correct correspondence between virtual and physical textures for test 1.

| Virtual Texture | | Physical Texture |
|---|---|---|
| Canvas A | ⇔ | Cardboard texture |
| Canvas B | ⇔ | Tiles texture |
| Canvas C | ⇔ | Phone case texture |
| Canvas D | ⇔ | Cork texture |

Table 2 presents the percentage of correct matches for each canvas-object correspondence.

**Table 2.** Percentage of correct correspondence between virtual and physical textures for test 1 according to volunteers.

|            | Carboard | Tiles | Phone Case | Cork | NONE |
|------------|----------|-------|------------|------|------|
| Canvas A   | 55%      | 0%    | 20%        | 10%  | 15%  |
| Canvas B   | 10%      | 45%   | 25%        | 15%  | 5%   |
| Canvas C   | 30%      | 25%   | 35%        | 5%   | 5%   |
| Canvas D   | 0%       | 20%   | 10%        | 70%  | 0%   |

The first conclusion is that each texture was correctly identified by most of the subjects. In detail, cork and cardboard are the textures with a higher correspondence rate with 70% and 55% correct correspondences, respectively, which suggests that uniform and regular textures are the ones with a better representation of the TT haptic device. The other two textures (tiles and phone case) that gathered fewer votes were perceived with greater difficulty by the users. In the opinion of volunteers, the phone case texture was hard to identify due to its irregularity, suggesting that more complex textures like this one are harder to reproduce in a TT haptic environment. Regarding the tile texture, there is no apparent reason for not being well recognized. A possible reason for this could be that the texture needs to be improved.

On canvas B, the texture of the tiles was sometimes mistaken with the phone case texture, while the opposite was also true, showing consistency in the results, although on canvas C the phone case texture was mistaken by some users with the cardboard texture because of its lines.

Overall, as seen in Table 2, the texture with more correct physical-virtual correspondences was cork, then cardboard texture in second place, then tiles, and finally the phone case. The texture that volunteers mentioned to be the easiest to identify was the cardboard, as shown in Figure 6, which highlights volunteers' preference for simpler textures over more complex ones.

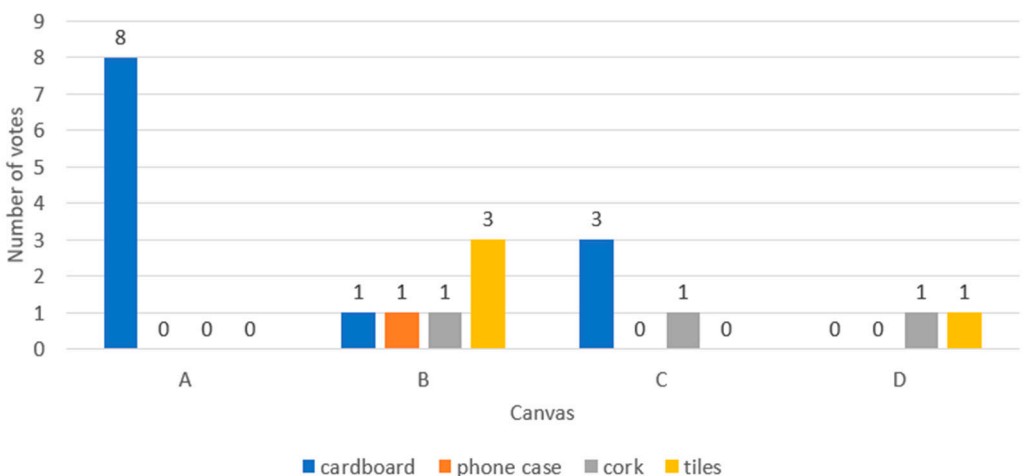

**Figure 6.** Easiest physical-virtual texture correspondence according to volunteers for test 1.

From the second test on, the main goal was to understand which of the virtual textures, volunteers select as the most adequate one to represent the physical texture of one of the objects from Figure 1.

For test 2, volunteers selected the texture that they thought was represented on the touchscreen as follows in Figure 7a.

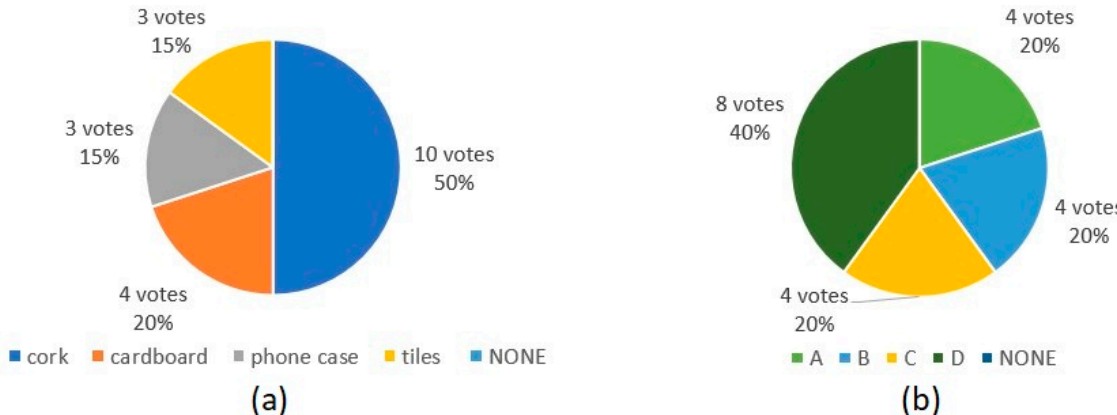

**Figure 7.** (**a**) Volunteers' perception of the virtual texture displayed on the touchscreen for test 2. (**b**) Volunteers' perception about the canvas that better represented the physical texture for test 2. According to survey questions from Appendix A.

Half of the volunteers (10 individuals) correctly identified the texture represented on the TT screen, which was cork. Next, volunteers were asked which virtual texture of the available canvasses (A, B, C, D, or NONE of them) better represented the physical texture of the object that was being reproduced, and the results are presented in Figure 7b. The most voted answer was canvas D with 40% of the votes (eight individuals out of 20), which corresponds to the sharp filter version of the cork texture. With this test, it is noticeable that the cork texture was easy to identify by the volunteers. The main adjectives used to describe the textures from this test were 'rough' and 'harsh'.

For the third test, the virtual texture reproduced on the haptic touchscreen was the phone case texture. The volunteers of the experiment identified with 55% of the votes (11 people out of 20) the correct physical texture that was being displayed on the haptic touchscreen, as shown in Figure 8a.

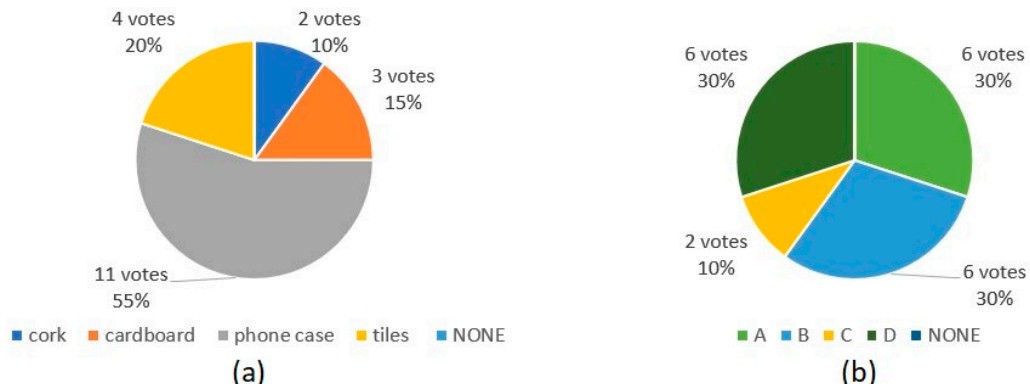

**Figure 8.** (**a**) Volunteers' perception of the virtual texture displayed on the touchscreen for test 3. (**b**) Volunteers' perception about the canvas that better represented the physical texture for test 3. According to survey questions from Appendix A.

When it comes to deciding which of the canvasses better represented the physical texture from the object, volunteers did not find a consensus, and canvasses A (mismatched image), B (sharp filter), and D (no texture) were tied at first place with 30% (6 individuals out 20) of the votes each as shown in Figure 8b). According to Figure 8, Canvasses A, B, and C are extremely similar in terms of texture. Canvas D was considered as a no texture canvas.

In the fourth test, the texture represented on the TT was the cardboard texture. The perception of volunteers about the virtual texture represented on this test is presented in Figure 9a.

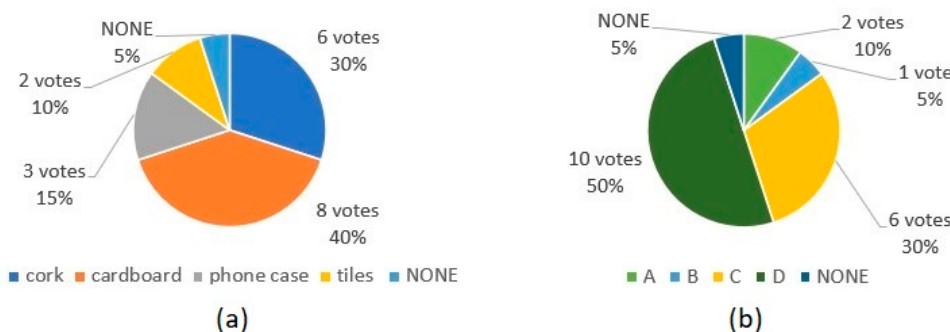

**Figure 9.** (**a**) Volunteers' perception of the virtual texture displayed on the touchscreen for test 4. (**b**) Volunteers' perception about the canvas that better represented the physical texture for test 4. According to survey questions from Appendix A.

It is noticeable that 40% (eight individuals out of 20) of users correctly identified the cardboard texture. Then, the volunteers were asked which of the canvasses better represented the physical texture during this test. Figure 9b presents the obtained results. The mismatched version of the cardboard texture (canvas D) was the most voted canvas. Many people used the word 'curvy' to describe the textures from this test.

The physical texture that was being represented in test 5 was the tiles. Volunteers were asked once again which virtual texture was being displayed on the touchscreen. The answers are presented in Figure 10a.

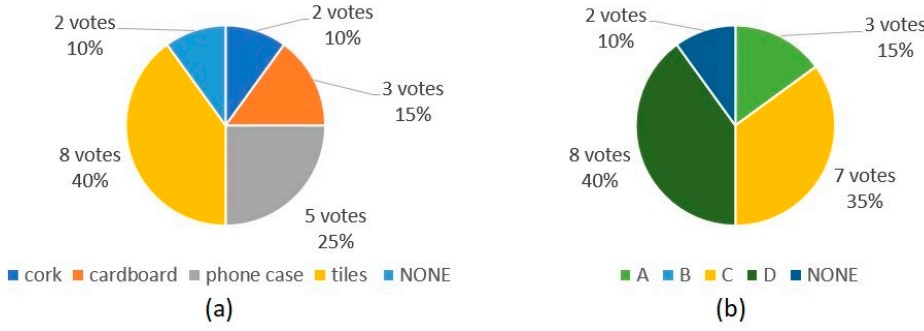

**Figure 10.** (**a**) Volunteers' perception of the virtual texture displayed on the touchscreen for test 5. (**b**) Volunteers' perception about the canvas that better represented the physical texture for test 5. According to survey questions from Appendix A.

Figure 10a reflects that eight out of 20 volunteers (40% of votes) considered correctly that the represented texture was the tiles, followed by the phone case texture that was voted by eight out of 20 individuals (25%).

In Figure 10b, 40% of the volunteers (eight out 20 participants in total) considered canvas A, which corresponds to the smoothed filtered version of the original texture, as the canvas that better represented the physical texture from this test. The second most voted option was canvas C, with seven out of 20 votes (35%), with only one less vote than canvas A, where it is represented the mismatched version of the original texture of the tiles.

After the execution of the five tests and right before the end of the survey that volunteers answered during the experiment (Appendix A), participants of the blind tests were inquired which of the tests (excluding test 1) was the easiest to identify the physical texture that was being reproduced on the haptic touchscreen. Figure 11a presents the obtained responses.

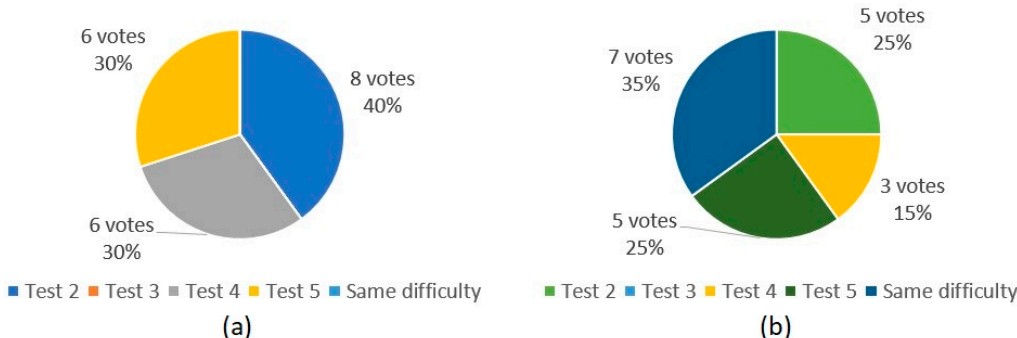

**Figure 11.** (**a**) Answers of the volunteers when questioned about which test was the easiest to identify the virtual texture on the touchscreen (tests 2 to 5). (**b**) Responses of the volunteers when inquired about which of the tests 2 to 5 was the easiest to describe the textures in it. According to survey questions from Appendix A.

The most popular answer was test 2, which corresponds to cork virtual texture, with a total of eight of the 20 votes (40%). With six of the 20 votes (30%) each, both tests 4 (cardboard texture) and 5 (tiles texture) were in the second position. Based on the previous distribution of votes, it is reasonable to say that users tend to find it easier to recognize virtual textures that have a uniform pattern (like cork) or a regular texture, such as cardboard or tiles textures, that occupy the second position with more votes. The phone case texture that has an irregular pattern was not even considered by a single volunteer in the answer to this question.

The second final question was designed to understand if any of the virtual textures from tests 2–5, was especially easy to describe its texture from the remaining tests. Figure 11b shows the obtained results. Most of the volunteers, seven out of 20 (35%), answered that they experienced the same difficulty performing this task in all tests from tests 2–5.

It is visible in the previous Table 3 that most of the virtual textures have a negative variation in the number of votes from test 1 to the remaining tests. All textures attain a higher number of votes during the first test (with exception of the phone case texture), where all virtual textures were reproduced at the same time, then the votes that were obtained individually during the remaining tests, where there was a single texture displayed at each time. This means that perception is sensitive to variations in the texture, as shown in the different variations of the same texture presented by the blurred filter, sharp filter, and the mismatched texture effect seen in tests 2–5.

**Table 3.** Comparison of volunteers' votes between test 1 and the remaining tests.

| Virtual Texture | Votes in Test 1 vs. Votes in Other Tests | Variation of Votes (%) |
|---|---|---|
| Cork | 14 (test 1) vs. 10 (test 2) | −20% |
| Phone case | 7 (test 1) vs. 11 (test 3) | +20% |
| Cardboard | 11 (test 1) vs. 8 (test 4) | −15% |
| Tiles | 9 (test 1) vs. 8 (test 5) | −5% |

Furthermore, it is interesting to observe that the object that obtained the worse result in test 1 (phone case) with only 35% of votes in Table 2 is the object with the higher number of votes in the other tests, with 55% of votes in test 3 (11 people out of 20) as seen in Figure 8a, while cork got 50% of votes (10 out of 20 volunteers) in Figure 7a, which was the score of the cork texture in test 2. Hence, the phone case texture benefits from not being compared with the remaining textures at the same time.

Adding to the previous results, three people referred that finger humidity and temperature, along with natural sweating, can affect users' perception of the reproduced textures on the haptic device. Two individuals mentioned that adding images (the visual component) to the haptic textures plays a decisive role when corresponding virtual textures to the

physical textures of a certain object and one person said that the user's ability to recognize virtual textures on electroadhesion-based haptic technology is improved with practice from test to test. Two volunteers considered that the mismatched version had more texture intensity than the sharp version in one of the tests, which points out that the recognition of textures' intensity can still be a little bit misleading sometimes.

## 5. Conclusions

This paper presented different UX tests designed to understand how textures created on a TT are perceived as a representation of real-world objects. The results show that uniform and regular textures tend to obtain better results than other more complex textures. Nevertheless, the environment where these textures are experienced can influence the obtained outcome. Just recall the results obtained with the phone case texture mentioned at the end of the previous section. A relevant aspect to be mentioned is that users identify virtual textures better through relative comparison between them, since the number of votes for correct correspondences is, in most cases, much higher in test 1 than the isolated votes on correct correspondences from any of the remaining tests, as shown in Table 3.

Based on the feedback received during the execution of the tests, some subjective factors significantly influence the UX and its perception of virtual textures: finger humidity, temperature, and hand sweating.

Regarding textures design, it became clear that there is still a lot to do: improving the quality of specific textures, or even improving the technology itself. Volunteers were able to notice with ease the different textures' intensities.

Furthermore, it was not possible to identify any type of preference by the volunteers of this experiment about one of the texture variations in tests 2–5 (blurred, sharp, mismatched, and the texture less version of the original texture), since the most voted answer in each of these tests was always different: for test 2 it was the sharp version, for test 3 the mismatched, the sharp, and the no-texture versions were tied at the first place, for test 4 the mismatched version, and for test 5 it was the blurred version.

Overall, it is possible to recognize different textures and transmit the idea of a specific one with this haptic device. Nonetheless, reproduction on the screen is not very realistic yet. This haptic device allows for identifying simple, regular, and homogeneous textures, but there are still several open challenges regarding the exploration of this technology and its relationship with human touch and sensibility.

The participants of this UX experiment showed interest in this type of haptic technology and mentioned its capability to be applied in many different areas. There is a consensus that this technology has a huge potential, but it was frequently mentioned by the volunteers that it is still a bit limited, and the sensibility of the human finger has not been sufficiently explored yet.

**Author Contributions:** Conceptualization, A.C. and P.C.; methodology, A.C. and P.C.; software, A.C.; validation, A.C.; formal analysis, A.C. and P.C.; investigation, A.C. and P.C.; resources, A.C. and P.C.; data curation, A.C.; writing—original draft preparation, A.C.; writing—review and editing, A.C. and P.C.; visualization, A.C.; supervision, P.C.; project administration, A.C. and P.C.; funding acquisition, P.C. All authors have read and agreed to the published version of the manuscript.

**Funding:** This work has been supported by FCT—Fundação para a Ciência e Tecnologia within the R&D Units Project Scope: UIDB/00319/2020.

**Informed Consent Statement:** Informed consent was obtained from all subjects involved in the study.

**Conflicts of Interest:** The authors declare no conflict of interest.

## Appendix A. Feedback Survey

**TEST 1**

**T1Q1** Establish the correspondence between each canvas and the textures from the objects:

|          | Cardboard | Phone case | Cork | Tiles | NONE |
|----------|-----------|-----------|------|-------|------|
| Canvas A |           |           |      |       |      |
| Canvas B |           |           |      |       |      |
| Canvas C |           |           |      |       |      |
| Canvas D |           |           |      |       |      |

**T1Q2** Which of the correspondences was the easiest to identify? Select only one.

|          | Cardboard | Phone case | Cork | Tiles |
|----------|-----------|-----------|------|-------|
| Canvas A |           |           |      |       |
| Canvas B |           |           |      |       |
| Canvas C |           |           |      |       |
| Canvas D |           |           |      |       |

**T1Q3.1** Were there any objects without correspondence? (Yes/No).
**T1Q3.2 (OPTIONAL)** If yes, please identify the object.

**TEST 2**

**T2Q1** Which of the textures from the presented objects is being represented on the haptic touchscreen? ('NONE' is a valid answer)
**T2Q2.1** Which of the canvasses (A, B, C or D) better represents the texture from that object? ('NONE' is a valid answer)
**T2Q2.2** How would you describe the texture of the remaining canvasses? This is an open question.
**T2Q3** Please order the canvasses according to their texture intensity:

|                  | Canvas A | Canvas B | Canvas C | Canvas D |
|------------------|----------|----------|----------|----------|
| 1st most intense |          |          |          |          |
| 2nd most intense |          |          |          |          |
| 3rd most intense |          |          |          |          |
| 4th most intense |          |          |          |          |

**TEST 3**

**T3Q1** Which of the textures from the presented objects is being represented on the haptic touchscreen? ('NONE' is a valid answer)
**T3Q2.1** Which of the canvasses (A, B, C or D) better represents the texture from that object? ('NONE' is a valid answer)
**T3Q2.2** How would you describe the texture of the remaining canvasses? This is an open question.
**T3Q3** Please order the canvasses according to their texture intensity:

|                  | Canvas A | Canvas B | Canvas C | Canvas D |
|------------------|----------|----------|----------|----------|
| 1st most intense |          |          |          |          |
| 2nd most intense |          |          |          |          |
| 3rd most intense |          |          |          |          |
| 4th most intense |          |          |          |          |

**TEST 4**

**T4Q1** Which of the textures from the presented objects is being represented on the haptic touchscreen? ('NONE' is a valid answer).
**T4Q2.1** Which of the canvasses (A, B, C or D) better represents the texture from that object? ('NONE' is a valid answer).

**T4Q2.2** How would you describe the texture of the remaining canvasses? This is an open question.
**T4Q3** Please order the canvasses according to their texture intensity:

|                  | Canvas A | Canvas B | Canvas C | Canvas D |
|------------------|----------|----------|----------|----------|
| 1st most intense |          |          |          |          |
| 2nd most intense |          |          |          |          |
| 3rd most intense |          |          |          |          |
| 4th most intense |          |          |          |          |

**TEST 5**

**T5Q1** Which of the textures from the presented objects is being represented on the haptic touchscreen? ('NONE' is a valid answer)
**T5Q2.1** Which of the canvasses (A, B, C or D) better represents the texture from that object? ('NONE' is a valid answer)
**T5Q2.2** How would you describe the texture of the remaining canvasses? This is an open question.
**T5Q3** Please order the canvasses according to their texture intensity:

|                  | Canvas A | Canvas B | Canvas C | Canvas D |
|------------------|----------|----------|----------|----------|
| 1st most intense |          |          |          |          |
| 2nd most intense |          |          |          |          |
| 3rd most intense |          |          |          |          |
| 4th most intense |          |          |          |          |

**FINAL QUESTIONS**

**FQ1** Which of the canvasses (A, B, C or D) was easier to identify the object's texture? ('Same difficulty' is a valid answer).
**FQ2** Which of the canvasses (A, B, C or D) was easier to describe the object's texture? ('Same difficulty' is a valid answer).
**FQ3 (OPTIONAL)** Feel free to add any other comments to this experiment.

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
