# Peer review of "Usability Tests for Texture Comparison in an Electroadhesion-Based Haptic Device"

_mti, doi:10.3390/mti6120108_

Round 1
Reviewer 1 Report
| The paper proposes a a set of experiments using TanvasTouch electroadhesion based haptic technology. The authors intend to contribute to the literature assessing how a texture created on a TanvasTouch device can be perceived as a representation of a real-world object. The practical application of simulations and HMI in general is a very relevant topic that has received quite a lot of attention recently both in academia and in practice. Haptic displays in particular have been gaining more relevance over the recent years. Although exists a lot of different papers in the area, this paper proposes a set of experiments. The research questions are clear, and the research method selected is appropriate for the type of research conducted. But the research background is not robust and has to be enlarged citing other papers that focuses on this topic, in order to expand the research foundation. The experiements description could be enlarged and the proposed approach could be adopted for other similar cases. Results need to be more clear in my opinion, showing how they answer the research questions. In my opinion, although the workflow is clear, but the paper structure could be improved and expanded. Good language and good academic way of writing. Probably the paper is slightly short by journal standards. In my opinion, the paper could be accepted but needs a major revision. The figures resolution could be improved and figures enlarged in order to enhance the clarity and the overall quality of the work. I suggest to follow the correct template of MTI MDPI journal. I want to encourage this work because has potential to be an interesting paper. |
Reviewer 2 Report
The paper presents a set of test to evaluate electroadhesion haptic TanvasTouch. The paper must be highly improved to be publicable in the journal in the next terms:
- The paper format does not follow the journal format at all. It seem to be a paper from a conference.
- The introduction must be highly improved showing deeply the environment of haptic technologies, where are they useful.
- The capabilities of TanvasTouch should be explained as it is a commercial product. Technical specification must be shown in order to know its capabilities and limitations
- In section II, each type of haptic displays must be more explained. Their physical characteristics and differences between them in terms of technology and applicability.
- Figures 3, 5, 6, 7, 8, 9, 10 and 12 are too small and the text is unreadable
- Figure 11 is missing (figure 12 is figure 11)
- In "D. Conclusions" it is mention some feedback received from users. This feedback, in my oppinion, should be commented in another section and not the conclusions.
- The paper seems to be more an evaluation of the product than a scientific study
Reviewer 3 Report
Dear Authors, please find my comments and suggestions below.
Please increase the font size of the text in the figures, especially in figures 7-12. This will improve their readability.
Figure 12 is numbered incorrectly. It should be 11. Please fix it.
Please don't use a colon at the end of a sentence like here
... a phone case, as presented in Figure 1:
and here
Volunteers elected the texture that they thought was being represented on the touchscreen as follows in Figure 7 a):
and so on.
Also, I would like to see more technical details of your research. What are the characteristics of the used device? You may describe the resolution, pixel density, and other characteristics specific to the used haptic display. What software and tools did you use? What are the texture parameters? I think you should add it to the Paper. This information may be interesting to some readers.
Round 2
Reviewer 1 Report
The paper is surely ehnanced since the last revision, in particular the state of art was enlarged in order to give an holistic comprehension of the technology and a more robust base to the research work. Despite these improvements, I think that some important things are missing: in introduction section, is important to define the aim of the paper, in the current manuscript the aim is only stated in the abstract and in line 199. Please move it to the end of introduction section. Moreover, some informations in introduction could be moved to state of art chapter, in introdution authors have just to give a general overview of the technologies and their limitations. The participants are not completely described: number, sex, age information are missing and is an important part in order to describe the context of the tests. The results section was improved and critically discussed. Good idea to add the appendix to show the survey. Minor format issue: the text has to be justified in all the sections.
Reviewer 2 Report
The paper has been highly improved. In fact, it has been almost rewriten. The experiments have been clearly explained as well as the obtained results.
Only a couple of suggestions:
- Don't start paragraphs using "also"
- In page 6 line 285 "begging" is begining"
